# Comparison of ERlangen Score with pTau/Aβ1-42 Ratio for Predicting Cognitive Decline and Conversion to Alzheimer’s Disease

**DOI:** 10.3390/brainsci15040334

**Published:** 2025-03-23

**Authors:** Julian Alexander Schwarz, Pauline Schulz, Janine Utz, Laura Rudtke, Johannes Jablonowski, Neele Klement, Piotr Lewczuk, Johannes Kornhuber, Juan Manuel Maler, Timo Jan Oberstein

**Affiliations:** 1Department of Psychiatry and Psychotherapy, Friedrich-Alexander-Universität Erlangen-Nürnberg, 91054 Erlangen, Germany; julian.schwarz@uk-erlangen.de (J.A.S.);; 2Department of Neurodegeneration Diagnostics, Medical University of Bialystok, 15-267 Białystok, Poland; 3Department of Biochemical Diagnostics, University Hospital of Bialystok, 15-267 Białystok, Poland

**Keywords:** ERlangen Score, risk of dementia, area under the curve, pTau/Aß1-42, comparison, cognitive decline

## Abstract

**Background/Objectives**: The ERlangen Score (ERS) and the pTau/Aβ1-42 ratio are dementia risk scores that use only surrogate markers of amyloid and tau pathology, whose performance has taken on added importance with the advent of anti-amyloid antibody therapies. Direct comparisons between the scores are limited, which is why the performance of the ERlangen Score (ERS) and the pTau/Aβ1-42 ratio in predicting cognitive decline and dementia risk were compared. **Methods**: Measurements of Aβ1-42, Aβ1-40, and pTau181 were conducted in cerebrospinal fluid samples using immunoassays. Linear mixed models and the area under the receiver operating characteristic curve (AUC, receiver operating characteristic = ROC) of 259 non-demented subjects were calculated. **Results**: The pTau/Aβ1-42 ratio correctly identified 55 out of 60 individuals with a positive Aβ1-42/Aβ1-40 ratio and pTau181 as having Alzheimer’s disease (AD), while the ERS correctly identified all of these individuals. The model using the ERS to predict cognitive trajectories (Akaike Information Criterion AIC = 2365) exhibited a marginally superior fit than the model using the pTau/Aβ1-42 ratio (AIC = 2371). There was no statistically significant difference in the AUC of the ERS (0.717) for dementia risk compared to the pTau/Aβ1-42 ratio (0.739), *p* = 0.179. However, when the Aβ1-42/Aβ1-40 ratio was not included in the ERS (AUC = 0.685), the pTau/Aβ1-42 score was found to be statistically significantly better, *p* = 0.007. **Conclusions**: The ERS showed an advantage in grouping, identifying all patients with a positive Aβ1-42/Aβ1-40 ratio and elevated pTau181 as having AD. The ERS and pTau/Aβ1-42 ratio were comparable in predicting dementia or cognitive decline. However, when the Aβ1-42/Aβ1-40 ratio is not available, the pTau/Aβ1-42 ratio should be preferred.

## 1. Introduction

The biological definition of Alzheimer’s disease (AD) relies mainly on surrogate indicators of amyloid plaques, neurofibrillary tangles, and neurodegeneration, with the aim of detecting and treating AD in its early stages [1,2,3,4]. These surrogate markers, including reduced levels of the 42-amino acid-long amyloid beta peptide (Aβ1-42) or elevated phosphorylated tau (pTau) and total tau (tTau) in cerebrospinal fluid, have been observed to change years before the onset of cognitive symptoms [5,6].

However, the individual parameters are contingent upon the utilized platform and inter- and intra-laboratory variation [1,2]. Consequently, there are no universally applicable threshold values. With regard to Aβ1-42, a normalization of values in relation to the 40-amino acid-long amyloid beta peptide (Aβ1-40) has been established, namely the Aβ1-42/Aβ1-40 ratio [3,4]. This improves the concordance with amyloid positron emission tomography (PET) imaging [7,8,9]. Additionally, previous reports indicated that a decline in the Aβ1-42/Aβ1-40 ratio may precede alterations in amyloid-PET imaging [9]. From a biological perspective, the ratio appears to be plausible, given that both peptides are generated by the same enzyme complex through a limited proteolytic process [10]. Nevertheless, the Aβ1-42/Aβ1-40 ratio does not address the question that arises when surrogate parameters for amyloidopathy, tauopathy, and neurodegeneration are subject to incongruous alterations. This further complicates the process of dementia risk assessment.

Previous reports have indicated that other ratios, such as the pTau/Aβ1-42 ratio and tTau/Aβ1-42 ratio, may offer advantages over the determination of individual parameters [7,11]. Additionally, fully standardized and automated devices have been observed to utilize these ratios, despite the difficulty in justifying these calculations from a biological perspective [12]. From a clinical perspective, these scores are a feasible option as they deliver a binary outcome, which evaluates amyloidopathy and tauopathy or amyloidopathy and neuronal injury simultaneously. This means that the optimal diagnostic sensitivity and specificity, as well as the predictive values, are achieved when both pathologies are present: increased pTau in the numerator and decreased Aß1-42 in the denominator, i.e., in cases which, from a biomarker perspective, clearly show an Alzheimer’s pathology. Furthermore, these scores ignore the more sensitive Aβ1-42/Aβ1-40 ratio and thresholds are not universally applicable.

The recently revised ERlangen Score (ERS) is an ordinal-scaled point score based on CSF Aβ1-42, Aβ1-40, pTau, and tTau with a range from zero to four [13]. Values for amyloidopathy (measured by decreased Aβ1-42) and tauopathy (measured by increased pTau181 or tTau) are categorized as normal (0), borderline (1), or definitely pathological (2). If pTau181 and tTau receive different classifications, the higher value is used for the tauopathy score. The amyloidopathy and tauopathy scores are then summed to obtain a total score ranging from zero to four. In a second step, the Aβ1-42/Aβ1-40 ratio is classified as normal (−1) or pathological (1) and added to the score. If the score falls outside this range, values of zero and four, respectively, are retained. Values above one are considered pathological. The ERS is defined in a manner that is not contingent on the specific thresholds employed in laboratory settings.

To date, there have been a number of reports that have misleadingly compared the pTau/Aβ1-42 ratio and the Aβ1-42/Aβ1-40 ratio with each other or their concordance with amyloid-PET imaging [7,8,12,14]. To our knowledge, there have been no direct comparisons in terms of dementia risk and cognitive decline conducted thus far.

The present study compares the area under the receiver operating characteristic curve (AUC) of the ERlangen Score and the pTau/Aβ1-42 ratio in order to identify individuals at risk of dementia. Additionally, it examines the ability of these instruments to predict cognitive decline.

## 2. Materials and Methods

### 2.1. Study Population

Individuals were recruited using consecutive sampling from the memory department of the clinic for psychiatry and psychotherapy at Friedrich-Alexander-University Erlangen-Nuremberg between April 2010 and June 2023. To be eligible, participants needed to have undergone a comprehensive cerebrospinal fluid (CSF) examination including Aβ1-42, the Aβ1-42/Aβ1-40 ratio, pTau181, and tTau, as well as a structural brain scan and a neuropsychological assessment within a 6-month timeframe. This study included individuals diagnosed with subjective cognitive impairment (SCI) or mild cognitive impairment (MCI) who were over the age of 50 and had at least one follow-up examination more than 12 months after the initial examination. Exclusion criteria included a history of stroke within the last six months, hallucinations, and inflammatory CNS diseases. Classification into SCI, MCI, or dementia was not based solely on Mini-Mental State Examination (MMSE) scores, but rather considered the full neuropsychological assessment, particularly in cases where MMSE scores were borderline [15]. The follow-up duration ranged from 12 to 156 months. The study protocol, with identification number 3987 and 19-426_1-B, was approved by the clinical ethics committee of the University of Erlangen-Nuremberg, and all participants provided written informed consent prior to their involvement in the study.

### 2.2. Neuropsychological Assessment

All participants in the Erlangen cohort were assessed using the German version of the Consortium to Establish a Registry for Alzheimer’s Disease neuropsychological battery plus (CERAD-NB+) [16].

### 2.3. CSF ELISA

The concentrations of Aβ1-40 and Aβ1-42 in CSF were determined with commercially available immunoassays from IBL International (Hamburg, Germany) and Fujirebio Europe (formerly Innogenetics, Ghent, Belgium). The Aβ immunoassays employed exclusively identify Aβ peptides commencing at Asp1, namely Aβ1-42 and Aβ1-40, respectively. Total tau and pTau181 levels in CSF were assessed using immunoassys provided by Fujirebio Europe and Innogenetics. When multiple pairs of values for Aβ 1-42 and pTau181 were available, IBL immunoassays or Fujrebio Innotest were preferred to Innogenetics Innotest. The calculation and values of the limits and ranges are described elsewhere [4,17]. Aβ1-42/Aβ1-40 ratio values were classified as pathologic using the Youden index for samples, measured using the automated Lumipulse^®^ Fujirebio platform if the values were below 0.06. A cut-off of 0.05 was set for all other immunoassays used.

The ERlangen Score has been computed as previously described [13]: In the event that the values for Aβ1-42 fall within a range of 10% below the specified cut-off value, a score of 1 is awarded. A score of 1 is assigned for both pTau181 and total tau in instances where the value is within 10% of the cut-off. In the event of a clearly pathological value (deviation > 10%), a score of 2 is assigned. In the case of both pTau181 and total tau, the higher score is selected and added to the score of Aβ1-42. In instances where the Aβ1-42/Aβ1-40 ratio has been determined, the sum score derived from the values of Aβ1-42, pTau181, and total tau is increased by one if the amyloid ratio is positive and the sum score is less than four, and decreased by one if the Aβ1-42/Aβ1-40 ratio is negative and the sum score is less than one. Values exceeding one are regarded as pathological. In instances where the ratio has not been established, the ordinal scale indicates the presence of low, medium, and high risk.

### 2.4. Statistics

The optimal thresholds for the pTau/Aβ1-42 ratio of the different immunoassays were determined using Youden’s index in conjunction with ROC analysis of neurochemically definite AD cases, i.e., pathological Aβ1-42/Aβ1-40, Aβ1-42, pTau181, and tau and biomarker-negative controls of all eligible individuals in the Erlangen cohort (*n* = 547). Cut-off values for the different immunoassay combinations are given in Appendix A.

The trajectories of the MMSE for the individual groups of the ERlangen Score, the dummy-coded groups for abnormal (0) and normal (1) ERSs, and the pTau/Aβ1-42 ratio were analyzed with linear mixed models. In mixed-effects modelling, the observations were considered to be at the lowest level (level 1) and nested at the subject level (level 2). The fixed effects were the follow-up time in months, the group, and the interaction between the group and the follow-up time in months. The models were constructed with a random intercept, defined as the subject’s identification number, and a slope, representing the time from baseline in years. An autoregressive covariance structure was employed for the repeated effects. The dependent variables were the z-values of MMSE tests utilized, which were calculated according to the Consortium to Establish a Registry for Alzheimer’s Disease (CERAD_Plus; German version, Memory Clinic Basel, 2002), taking into account age, sex, and educational level. Evaluation of demographically adjusted z-scores of the MMSE and other CERAD items using age, sex, and years of education as covariates has been described previously by Berres et al. using a predicted residual sum of squares statistic for model selection [18]. Maximum likelihood estimation (MLE) was used in the models with the general equation as follows:$$\begin{array}{c} {zMMSE}_{i,t}=\beta_{0}+\beta_{1}\left({Time}_{i,t}\right)+\beta_{2}\left({Group}_{i}\right)\\ \;\;\;\;\;\;\;\;\;\;\;\;\;\;\;\;\;\;\;\;\;\;\;\;\;\;\;\;\;\;+\beta_{3}\left({Time}_{i,t}\times {Group}_{i}\right)+\upsilon_{0i}\\ \;\;\;\;\;\;\;\;\;\;\;\;\;\;\;\;\;\;\;\;\;\;\;\;\;\;\;\;\;\;+\upsilon_{1i}\left({Time}_{i,t}\right)+\varepsilon_{i,t} \end{array}$$
where zMMSE is the MMSE z-score for subject *i* at time *t*, β_0_ is the fixed intercept (grand mean), β_1_ is the fixed effect of time (overall trend in MMSE decline), β_2_ is the fixed effect of group (difference between ERS groups), β_3_ is the interaction term (difference in MMSE decline rate between groups), u_i_ is the random intercept (individual differences at baseline), and ϵ_i,t_ is the residual error (individual variability over time). A total of 8 models were generated, in each of which the categorical variable group differed: ordinal scaled ERS classification, binary ERS classification, pTau/Aβ1-42 ratio classification, and ordinal scaled ERS without the Aβ1-42/Aβ1-40 ratio for non-demented individuals (SCI, MCI) as well as binary ERS classification and pTau/Aβ1-42 ratio classification for individuals with SCI and MCI. In addition to the Akaike Information Criterion (AIC) and the Bayesian Information Criterion (BIC), we used the conditional and marginal Root Mean Squared Error (RMSE) as a performance metric to measure the predictive accuracy of the model.$$RMSE=\sqrt{\frac{1}{n}\sum_{i=1}^{n}{\left(y_{i}-{\hat{y}}_{i}\right)}^{2}}$$
where y*_i_* is the observed MMSE z-score, ŷ*_i_* is the predicted MMSE z-score from the linear mixed model, and *n* is the number of observations.

To ascertain the accuracy of the ERlangen Score in comparison to the pTau/Aβ1-42 ratio, receiver operating characteristic (ROC) curves were constructed, including an area under the curve (AUC) analysis. To address the variability in follow-up times, a Cox proportional hazards model was used to estimate time-dependent probabilities for conversion to dementia. These predicted hazard probabilities were then used as test variables in the ROC analysis. The positive conditions were defined as worsening to dementia. The non-parametric method of DeLong et al. was used to compare the AUC in a paired-sample scenario [19].

Statistics analysis was performed using SPSS (version 28.0.1.1; SPSS, Chicago, IL, USA).

## 3. Results

A total of 259 individuals were eligible for inclusion in the study, comprising 104 females (40.2%) and 155 males (59.8%). A participant flow chart is provided in Appendix A for reference.

The mean age at baseline was 66.8 ± 9.3 years. The median MMSE score at baseline was twenty-seven with an interquartile range (IQR) of three. The median years of education was 13 (IQR = 5). The median follow-up period was 3.25 years (IQR = 4.9).

The largest group was that of the biomarker-negative control, comprising 87 individuals (33.6%). A clear AD biomarker constellation was observed in 30 individuals (11.6%), characterized by a pathological Aβ1-42, Aβ1-42/Aβ1-40 ratio, pTau181, and tTau. In all other cases, the biomarker profile was found to be incongruent. The group exhibiting normal Aβ1-42 and a pathological Aβ1-42/Aβ1-40 ratio, pTau181, and tTau comprised 58 individuals, representing 22.4% of the total sample. A total of 63 individuals (24.3%) exhibited pathological T and/or N values alongside a normal Aβ-42 and Aβ1-42/Aβ1-40 ratio. All other constellations were observed in less than 2% of cases. Table 1 shows how many subjects with the different biomarker constellations were classified as pathological according to the ERS or the pTau/Aβ1-42 ratio. The baseline demographics are presented in Table 2, Appendix A. Of the 259 non-demented participants, 81 developed dementia during the follow-up period.

### 3.1. Prediction Model of Cognitive Decline Using ERlangen Score or pTau/Aβ1-42 Ratio

To determine whether the ERS or the pTau/Aβ1-42 ratio was better at predicting cognitive function and its decline in terms of change in the MMSE, two linear mixed models were fitted with MMSE z scores as the dependent variable, and follow-up time in months and dummy-coded groups of the ERS or pTau/Aβ1-42 ratio were the independent variables. Using the Akaike Information Criterion (AIC) and the Bayesian Information Criterion (BIC), the pTau/Aβ1-42 ratio model (AIC = 2371.676, BIC = 2412.147) showed a better fit than the ERS model (AIC = 2375.271, BIC = 2442.723) as it uses less degrees of freedom. As soon as the five dimensions of the ERS were reduced to normal (ERS ≤ 1) and abnormal (ERS > 1), the underlying ERS model showed a best fit in both the AIC and BIC (AIC = 2364.685, BIC = 2405.156). Thus, in a scenario where only pathological and normal values are distinguished, with the ERS and pTau/Aβ1-42 ratio having an equal number of outcomes, the ERS model performs slightly better than the pTau/Aβ1-42 ratio model (Table 3). The assessment of model accuracy using the Marginal Root Mean Squared Error (RMSE) showed that both the ordinal-scaled and the binary version of the ERS predicted the fixed effects in the linear mixed model better than the pTau/Aβ1-42 ratio (Appendix A). Even when cognitive decline is considered separately for the MCI and SCI groups, the binominal version of the ERS performs as well as, or better than, the pTau/Aß1-42 ratio model (Appendix A). However, this is not the case when the Aβ1-42/Aβ1-40 ratio is not taken into account during the formation of the ERS (AIC = 2399.504, BIC = 2466.956, marginal RMSE = 1.798).

### 3.2. Discriminative Ability of ERS and pTau/Aβ1-42 Group for Dementia Risk

The receiver operating characteristic (ROC) curves for both the ERS and the pTau/Aβ1-42 ratio were examined to evaluate their capacity to predict the risk of conversion to dementia. The area under the curve (AUC) was 0.739 for the pTau/Aβ1-42 ratio and 0.717 for the ERS (Figure 1 and Appendix A). The analysis revealed no statistically significant difference in the predicted risk of dementia between the pTau/Aβ1-42 ratio and the ERS (*p* = 0.179). However, a statistically significant difference was observed between the pTau/Aβ1-42 ratio and the ERS when it was calculated without considering the Aβ1-42/Aβ1-40 ratio (AUC = 0.685, *p* = 0.007). The AUC values of the pTau/Aβ1-42 ratio and ERS for the various immunoassays employed are provided in Appendix A.

To compare the diagnostic odds ratios of the ERS and the pTau/Aβ1-42 ratio for predicting conversion to dementia, the optimal cut-offs for the pTau/Aβ1-42 ratio were determined using the Youden index. The diagnostic odds ratios did not differ significantly between the ERS [OR 5.729 95% CI (3.223, 10.151)] and pTau [OR 4.9583 95%CI (2.836, 8.757)]. A table showing sensitivity, specificity, positive, and negative likelihood ratios is provided in Appendix A.

## 4. Discussion

In a direct comparison, the performance of the pTau/Aβ1-42 ratio score, in terms of accuracy for predicting conversion to dementia in a cohort of 259 non-demented individuals, was similar to that of the ERlangen Score, and it was only slightly worse at predicting cognitive decline. Particularly, with the advent of biomarker-based therapies, such as anti-amyloid therapy, the question of how to select dementia scores is becoming increasingly important. In terms of dementia risk, the lack of information from the Aβ1-42/Aβ1-40 ratio appears to be largely compensated by the pTau/Aβ1-42 ratio. Using the pTau/Aβ1-42 ratio to group participants, 85.6% of those who had normal amyloid (A) markers but abnormal tau (T) or neuronal damage (N) markers were classified into the low-risk group for dementia. Consistent with this, ourselves and others have previously reported that there is no statistically significant increase in the risk of dementia in this group compared with the biomarker-negative group [17,20]. Others reported some cognitive decline compared with controls, but less than in AD [15]. However, using the pTau/Aβ1-42 ratio, five out of sixty participants with a positive Aβ1-42/Aβ1-40 ratio and pTau181 were not classified as having AD despite having probable AD according to the biomarker classification using NIA-AA and IWG-2 criteria [16,21,22]. In the light of emerging biomarker-based therapies, this misclassification by the pTau/Aβ1-42 ratio score should certainly be taken into account when compared with classification using a full set of CSF biomarkers, including the Aβ1-42/Aβ1-40 ratio. Consequently, in instances where the ratio has not been determined, the pTau/Aβ1-42 ratio is to be preferred. The ERlangen Score was designed to classify individuals, with a probable diagnosis of AD, based on CSF biomarkers as having a high risk of dementia, so no aberrant grouping was observed here when the Aβ1-42/Aβ1-40 ratio was included in the ERlangen Score calculation [13]. The ERS can also be determined independently of the Aβ1-42/Aβ1-40 ratio, but at the expense of its accuracy [13]. Accordingly, the present study showed that when the Aβ1-42/Aβ1-40 ratio is not taken into account when calculating the ERS, the pTau/Aβ1-42 ratio has a better model fit and a significantly higher AUC for the MMSE and the risk of dementia. In cohorts in whom Aβ1-40 has not been regularly determined, e.g., cohorts from the Alzheimer’s disease Neuroimaging Initiative [23], this study suggests that pTau/Aβ1-42 ratio score should be preferred. However, a disadvantage of the pTau/Aβ1-42 ratio is that the individual laboratory reference values are not comparable between methods and between centers [1,2,24].

The ERlangen Score combines the two neuropathological hallmarks of AD, amyloid pathology and tau pathology, in an ordinal score that allows the risk of dementia to be assessed independently of the laboratory and does not, like the pTau/Aβ1-42, assume that the biological processes of amyloid and tau deposits are in a linear relationship with each other [13,25].

However, it needs to be stressed that in contrast to the Aβ1-42/Aβ1-40ratio, which is a measure for the normalization of Aβ1-42 for the total CSF amount of Aβ (of which Aβ1-40 is the most abundant isoform), the pTau/Aβ1-42 ratio attempts to normalize a biomarker of one pathophysiological process (amyloidosis) for a biomarker of another process (neurodegeneration). Furthermore, any quotient, by its mathematical definition, explicitly assumes that the relation of the two quantities (here, a biomarker of amyloidosis and a biomarker of neurodegeneration) is linear. The pTau/Aβ1-42 ratio provides a more reliable interpretation of the AD biomarkers when the Aβ1-42/Aβ1-40 ratio is not available, particularly in clear cases [26]. However, it should be considered a purely mechanistic concept [27].

## 5. Limitations

Due to the continuous enrolment of subjects, the possible follow-up time of each subject varied, which, together with the loss to follow-up and the presumed heterogeneous dementia risk of subjects with subjective and mild cognitive impairment, explains the less than 80% accuracy in predicting dementia risk. In addition, it is not possible to draw any conclusions regarding the comparative performance of the various immunoassays used, given that they were not applied randomly over the entire observation period. The fact that the cohort was mainly composed of white Caucasians living in Germany means that it has limited applicability to other regions of the world and other origins. In addition, the cohort consisted mainly of well-educated individuals and may not accurately represent the general population. Nevertheless, given that the patients constituting the study cohort are treated in memory outpatient clinics and exhibit a higher level of compliance than is typical, the results may be considered representative of patients who may be eligible for anti-amyloid therapy. In terms of the biomarker profiles and MMSE values, the cohort shares characteristics with cohorts from other regions of the world [28,29].

## 6. Conclusions

The ERlangen and pTau/Aβ1-42 score showed no significant difference in accuracy concerning the prediction of dementia risk or cognitive decline. However, when the Aβ1-42/Aβ1-40 ratio is not available, the pTau/Aβ1-42 score should be preferred. Further investigation is required to ascertain the significance of ATN-classification-deviating allocation as determined by the pTau/Aβ1-42 score.

## Figures and Tables

**Figure 1 brainsci-15-00334-f001:**
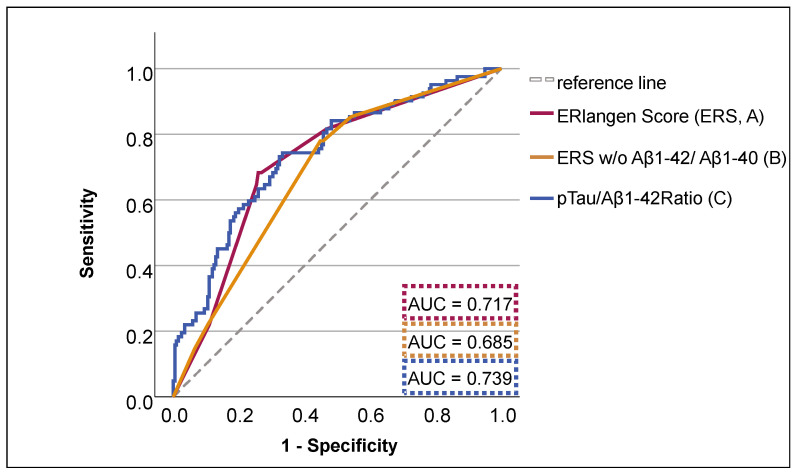
The receiver operating characteristic (ROC) curve is utilized for the prediction of dementia, employing the ERlangen Score (ERS) (A), the pTau/Aβ42 ratio (B), and the ERS excluding the Aβ1-42/Aβ1-40 ratio (C). The hazard probabilities are predicted from a Cox regression model, which was specified for a 10-year follow-up period. AUC = area under the curve.

**Table 1 brainsci-15-00334-t001:** Different biomarker constellations were classified as low or high dementia risk according to the pTau/Aβ1-42 ratio or the ERlangen Score (ERS). These constellations included the Aβ1-42/Aβ1-40 ratio (R), Aβ1-42 (A), pTau181 (T), and total Tau (N) in the CSF.

			Normal Biomarkers	Alzheimer’s Continuum									SNAP^1^			
			R–^2^A–T−N–	R+^3^A–T–N–	R–A+T–N–	R+A+T–N–	R+A–T–N+	R+A+T–N+	R+A–T+N–	R+A+T+N–	R+A–T+N+	R+A+T+N+	R–A–T–N+	R–A–T+N–	R–A–T+N+	Total
pTau/Aβ1-42 ratio	Low		87	1	5	2	3	0	1	0	4	0	9	16	29	157
	High		0	0	0	3	0	2	1	3	54	30	0	2	7	102
ERS	Low	0	87	0	4	0	0	0	0	0	0	0	4	9	3	107
		1	0	1	1	0	0	0	0	0	0	0	5	9	33	49
	High^4^	2	0	0	0	1	2	0	1	0	0	0	0	0	0	4
		3	0	0	0	4	1	0	1	0	57	0	0	0	0	63
		4	0	0	0	0	0	2	0	3	1	30	0	0	0	36

^1^ SNAP = suspected non-Alzheimer pathology, ^2^ “+” = pathological, ^3^ “−“ = normal, and ^4^ ERS ≥ 2 is considered to be suspicious for Alzheimer’s disease.

**Table 2 brainsci-15-00334-t002:** Baseline characteristics of individuals classified as low risk and high risk according to the pTau/Aβ1-42 ratio score and ERS categories. Error! Not a valid link.SEM = standard error of means.

	pTau/Aβ1-42 Ratio								ERS							
	Low				High				Low				High			
	*n*		%		*n*		%		*n*		%		*n*		%	
Total (female)	157	(55)	61	(21)	102	(49)	39	(19)	156	(56)	60	(22)	103	(48)	40	(19)
Total (MCI)	117	(40)	56	(47)	91	(45)	44	(53)	117	(39)	56	(46)	91	(46)	44	(54)
Total (SCI)	40	(16)	78	(84)	11	(3)	22	(16)	39	(16)	76	(84)	12	(3)	24	(16)
	mean		*SEM*		mean		*SEM*		mean		*SEM*		mean		*SEM*	
Age [years]	63.5		0.7		71.8		0.8		63.3		0.7		72.0		0.8	
Education [years]	14.3		0.2		13.2		0.3		14.3		0.2		13.3		0.3	
MMSE	27.4		0.2		26.0		0.2		27.3		0.2		26.2		0.2	
Follow-Up [years]	4.9		0.3		3.3		0.2		4.8		0.3		3.5		0.2	

**Table 3 brainsci-15-00334-t003:** Linear mixed model fits and estimates for comparison of MMSE z scores and their trajectories between normal and pathological ERSs (A) and pTau/Aβ1-42 ratio (B) with respective non-pathological group as reference.

A	Fixed Effects						B	Fixed Effects					
	Estimate	*SEM*	*t*	*p*	95% CI			Estimate	*SEM*	*t*	*p*	95% CI	
Intercept	−1.30	0.09	−14.79	<0.001	−1.47	−1.12	Intercept	−1.32	0.09	−15.25	<0.001	−1.49	−1.15
Follow-up [Y]	−0.07	0.03	−2.31	0.028	−0.17	−0.01	Follow-up [Y]	−0.11	0.04	−2.85	0.008	−0.19	−0.03
Pathologic ERS	−0.40	0.15	−2.45	0.015	−0.65	−0.07	Pathologic pTau/Aβ1−42 ratio	−0.30	0.15	−1.99	0.048	−0.60	0.00
Pathologic ERS × Follow-up [Y]	−0.40	0.06	−6.13	<0.001	−0.57	−0.29	Pathologic pTau/Aβ1−42 ratio × follow-up [Y]	−0.44	0.08	−5.80	<0.001	−0.59	−0.29
Model fit							Model fit						
AIC	2346.69						AIC	2371.68					
BIC	2405.16						BIC	2412.15					

## Data Availability

The data supporting the findings of this study are available within the article and/or its Appendix A. Further datasets generated and/or analyzed in the current study are available from the corresponding author upon reasonable request.

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
