# Peer review of "Comparison of ERlangen Score with pTau/Aβ1-42 Ratio for Predicting Cognitive Decline and Conversion to Alzheimer’s Disease"

_brainsci, 2025, doi:10.3390/brainsci15040334_

Round 1

Reviewer 1 Report

Comments and Suggestions for Authors

Minor revision.

Although the sample size of the study was small, the conclusions have very important implications for clinical diagnosis. If the same conclusion can be drawn in blood indicators, the study may be more significant and more guiding for clinical diagnosis.

Recommendation:

The authors are requested to provide the ROC and AUC in this study.

Comments on the Quality of English Language

Good

Author Response

Comments 1: Although the sample size of the study was small, the conclusions have very important implications for clinical diagnosis. If the same conclusion can be drawn in blood indicators, the study may be more significant and more guiding for clinical diagnosis.

Answer 1: We would like to express our gratitude for your commentary. The CSF biomarkers were determined in a laboratory certified according to DIN EN ISO 15189. However, it should be noted that, in contrast to CSF markers, the validation of blood biomarkers for routine diagnostic use has not yet been sufficiently established. The objective of the present study was to make a comparison between the scores that are already in routine use. 

Comment 2: The authors are requested to provide the ROC and AUC in this study.

Answer 2: As requested, we have added Figure 1 to show the ROC and AUC graphically.

Reviewer 2 Report

Comments and Suggestions for Authors

The work is interesting and can be published with some changes.

Point 1

On lines 151-152:

What relationship exists between the variables, sex, age and level of education.

Explain a little more about the 8 models.

Point 2

On lines 162,163

Explain the sampling technique used to collect data for the study.

Point 3

On lines 185-187

The authors use the AIC and BIC criteria for the prediction capacity of the model, it is not very advisable, it is necessary to use other metrics that help to select the best model. In prediction models it is vital to use other metrics to determine the quality of prediction, for example MAPE (Mean Absolute Percentage Error), it is a statistical measure that calculates the precision of forecasts.

The authors must add other criteria for model validation.

Point 4

On line 328.

Limitations: they present the disadvantages of their work. However, the question is: Under what conditions can it be applied to other regions of the world.

Author Response

Comments 1: On lines 151-152: What relationship exists between the variables, sex, age and level of education.

Explain a little more about the 8 models.

Answer 1: We have amended the text as follows: „Evaluation of demographically adjusted z-scores of the MMSE and other CERAD items using age, sex and years of education as covariates has been described previously by Berres et al. using a predicted residual sum of squares statistic for model selection.“ lines 155-158 and

"Maximum likelihood estimation (MLE) was used in the models with the general equation:

〖zMMSE〗_(i,t)=β_0+ β_1  (〖Time〗_(i,t) )+ β_2 (〖Group〗_i )+ β_3 (〖Time〗_(i,t)  x 〖Group〗_i )+ υ_i+ ε_(i,t)

where zMMSE is the MMSE z-score for subject i at time t, β0 is the fixed intercept (grand mean), β1 is the fixed effect of time (overall trend in MMSE decline), β2 is the fixed effect of group (difference between ERS groups), β3 ist the interaction term (difference in MMSE decline rate between groups), ui is the random intercept (individual differences at baseline), and ϵi,t is the residual error (individual variability over time). A total of 8 models were generated, in each of which the categorical variable group differed: ordinal scaled ERS classification, binary ERS classification, pTau/Aβ1-42 ratio classifi-cation and ordinal scaled ERS without the Aβ1-42/ Aβ1-40 ratio for non-demented individuals (SCI, MCI) as well as binary ERS classification and pTau/Aβ1-42 ratio classification for individuals with SCI and MCI." lines 158-170

Comments 2: On lines 162,163 Explain the sampling technique used to collect data for the study

Answer 2: 

We updated the manuscript as follows: “Individuals were recruited using consecutive sampling from the memory department of the clinic for psychiatry and psychotherapy at Friedrich-Alexander-University Erlangen-Nuremberg between April 2010 and June 2023.” lines 93-95

“To address the variability in follow-up times, a Cox proportional hazards model was used to estimate time-dependent probabilities for conversion to dementia. These predicted hazard probabilities were then used as test variables in the ROC analysis.” lines 177-180

The results were adjusted accordingly and shown in Figure 1 and A2 (see Reviewer 1)

Comments 3: On lines 185-187 The authors use the AIC and BIC criteria for the prediction capacity of the model, it is not very advisable, it is necessary to use other metrics that help to select the best model. In prediction models it is vital to use other metrics to determine the quality of prediction, for example MAPE (Mean Absolute Percentage Error), it is a statistical measure that calculates the precision of forecasts. The authors must add other criteria for model validation.

Answer 3: 

We are pleased to take up the reviewer's suggestion and have added the root mean square error (RMSE) to the text as a measure of model accuracy, as we had too many values equal to zero due to the use of the z-score of the MMSE as the dependent variable to use the MAPE. lines 171ff

The results were adjusted accordingly.

Comments 4: On line 328. Limitations: they present the disadvantages of their work. However, the question is: Under what conditions can it be applied to other regions of the world.

Answer 4: 

We are pleased to implement this suggestion as follows: „Nevertheless, given that the patients constituting the study cohort are treated in memory outpatient clinics and exhibit a higher level of compliance than is typical, the results may be considered representative of patients who may be eligible for anti-amyloid therapy. In terms of biomarker profile and MMSE values, it shares characteristics to cohorts from other regions of the world [27, 28].” lines 385-389

  1. Fowler, Christopher, et al. "Fifteen years of the Australian Imaging, Biomarkers and Lifestyle (AIBL) study: progress and observations from 2,359 older adults spanning the spectrum from cognitive normality to Alzheimer’s disease." Journal of Alzheimer's disease reports 5.1 (2021): 443-468.
  2. Potashman, Michele, et al. "Estimating progression rates across the spectrum of Alzheimer’s disease for amyloid-positive individuals using national Alzheimer’s coordinating center data." Neurology and Therapy 10.2 (2021): 941-953

Reviewer 3 Report

Comments and Suggestions for Authors

Reviewer’s comment for Manuscript - brainsci-3496674

This is an interesting piece of statistical study. However, some improvement in data presentations is required.

  1. Line 71-73 – What exact value of Aβ1-42 and pTau was considered for 0, 1, and 2.
  2. In none of the tables, the exact Aβ1-42/ Aβ1-40 ratio are mentioned. What is the exact value and how it was normalized?
  3. Line 19-21 – How this data was measured? What is the upper limit and lower limit of that value that was obtained? The authors must provide the Raw data of immunoassay for different groups (ERS 0 to 4) in the SI.
  4. Apart from CSF markers, there are certain blood biomarkers for AD and/or dementia (like Aβ43, Flotilin, B-FABP, etc). But these were not analyzed at all why?
  5. In line 22-24 – ‘The pTau/Aβ1-42 ratio correctly identified 55 out of 60 individuals with a positive Aβ 1-42/Aβ 1-40 ratio and pTau181 as having Alzheimer's disease (AD), while the ERS correctly identified all of these individuals’ – What is the ERS score of the first 55 individuals? Is it mentioned in Table 1?
  6. There is some confusion regarding the tables that were submitted in the Main manuscript and SI. Those tables should only be submitted in SI which are NOT there in the main manuscript.
  7. What value of AIC and BIC should be considered above baseline, moderate or high? The authors have not given any classification on those values or value ranges.
  8. If the Aβ1-42/ Aβ1-40 are produced from the same peptide by proteolytic cleavage, how come their ratio is different? In principle, the change in concentration of both peptides should be the same. So, their ratio should not Change.
  9. In the CSF immunoassay, what is the ratio of Tau/pTau level? Does it indicate any relation with dementia which can be linked with ERS for combined prediction?
  10. All the abbreviations (AIC, AUC, MMSE, etc) should be mentioned first when the term appears in the manuscript and not randomly elsewhere
  11. Since the samples were obtained from clinical studies the PET imaging data must be available for most of the samples. Can that data be correlated with the ERS score or Aβ 1-42/1-40 ratio?

Author Response

Comments 1: Line 71-73 – What exact value of Aβ1-42 and pTau was considered for 0, 1, and 2.

Response 1: We kindly refer to lines 122-123: “The calculation and the values of the limits and ranges are described elsewhere [4, 14]” and lines 124-131: “In the event that the values for Aβ1-42 fall within a range of 10% below the specified cut-off value, a score of 1 is awarded. A score of 1 is assigned for both pTau181 and to-tal tau in instances where the value is within 10% of the cut-off. In the event of a clear-ly pathological value (deviation >10%), a score of 2 is assigned. In the case of both pTau181 and total tau, the higher score is selected and added to the score of Aβ1-42.”

Lewczuk, P., Esselmann, H., Otto, M., Maler, J. M., Henkel, A. W., Henkel, M. K., ... & Wiltfang, J. (2004). Neurochemical diagnosis of Alzheimer’s dementia by CSF Aβ42, Aβ42/Aβ40 ratio and total tau. Neurobiology of aging, 25(3), 273-281

Oberstein, T. J., Schmidt, M. A., Florvaag, A., Haas, A. L., Siegmann, E. M., Olm, P., ... & Maler, J. M. (2022). Amyloid-β levels and cognitive trajectories in non-demented pTau181-positive subjects without amyloidopathy. Brain, 145(11), 4032-4041.

Comments 2: In none of the tables, the exact Aβ1-42/ Aβ1-40 ratio are mentioned. What is the exact value and how it was normalized?

Response 2: The manuscript has been amended as follows: “Aβ1-42/ Aβ1-40 ratio values were classified as pathologic using the Youden index for samples measured using the automated Lumipulse® Fujirebio platform if the values were below 0.06. A cut-off of 0.05 was set for all other immunoassays used. (lines 123-125)”

Comments 3 Line 19-21 – How this data was measured? What is the upper limit and lower limit of that value that was obtained? The authors must provide the Raw data of immunoassay for different groups (ERS 0 to 4) in the SI.

Response 3: Please refer to tables A1 and A3 as well as lines 122-123

Lewczuk, P., Esselmann, H., Otto, M., Maler, J. M., Henkel, A. W., Henkel, M. K., ... & Wiltfang, J. (2004). Neurochemical diagnosis of Alzheimer’s dementia by CSF Aβ42, Aβ42/Aβ40 ratio and total tau. Neurobiology of aging, 25(3), 273-281

Oberstein, T. J., Schmidt, M. A., Florvaag, A., Haas, A. L., Siegmann, E. M., Olm, P., ... & Maler, J. M. (2022). Amyloid-β levels and cognitive trajectories in non-demented pTau181-positive subjects without amyloidopathy. Brain, 145(11), 4032-4041.

Mean and SEM of CSF Aβ1-42/ Aβ1-40 ratio, Aβ1-42, and pTau181 for the different immunoassays is now provided in Table A2

Comments 4: Apart from CSF markers, there are certain blood biomarkers for AD and/or dementia (like Aβ43, Flotilin, B-FABP, etc). But these were not analyzed at all why?

Response 4: Please refer to our response to reviewer 1.

Comments 5: In line 22-24 – ‘The pTau/Aβ1-42 ratio correctly identified 55 out of 60 individuals with a positive Aβ 1-42/Aβ 1-40 ratio and pTau181 as having Alzheimer's disease (AD), while the ERS correctly identified all of these individuals’ – What is the ERS score of the first 55 individuals? Is it mentioned in Table 1?

Response 5: The reviewer is right that it can be deducted from table 1 (1 with an ERS of 4, 58 with an ERS of 3 and 1 with an ERS of 2).

Comments 6: There is some confusion regarding the tables that were submitted in the Main manuscript and SI. Those tables should only be submitted in SI which are NOT there in the main manuscript.

Response 6: Thank you for the note, we submitted SI separately from the main manuscript.

Comments 7: What value of AIC and BIC should be considered above baseline, moderate or high? The authors have not given any classification on those values or value ranges.

Response 7: We would politely suggest that you refer to the answer provided to Reviewer 2.

Comments 8: If the Aβ1-42/ Aβ1-40 are produced from the same peptide by proteolytic cleavage, how come their ratio is different? In principle, the change in concentration of both peptides should be the same. So, their ratio should not Change.

Response 8: Answer: We are very grateful for this commentary, which once again highlights the importance of the Ab42/Ab40 ratio. The Aβ1-42/ Aβ1-40 ratio can be influenced by many factors, one of which is changes in the preferential cleavage of the g-secretase multi-enzyme complex, e.g. see Haass C, De Strooper B. The presenilins in Alzheimer's disease--proteolysis holds the key. Science. 1999 Oct 29;286(5441):916-9. doi: 10.1126/science.286.5441.916. PMID: 10542139.

The benefits of the Aβ1-42/ Aβ1-40 ratio have been reviewed some time ago, see for example Hansson, O., Lehmann, S., Otto, M. et al. Advantages and disadvantages of the use of the CSF Amyloid β (Aβ) 42/40 ratio in the diagnosis of Alzheimer’s Disease. Alz Res Therapy 11, 34 (2019). https://doi.org/10.1186/s13195-019-0485-0

Prof. Lewczuk was so kind to amend the following passage to the current manuscript: “However, it needs to be stressed that in contrast to Aβ1-42/1-40 ratio, which is a measure for normalization of Aβ1-42 for the total CSF amount of Aβ (of which Aβ1-40 is the most abundant isoform), pTau/Aβ1-42 attempts to normalize a biomarker of one pathophysiological process (amyloidosis) for a biomarker of another process (neurodegeneration). Furthermore, any quotient, by its mathematical definition, explicitly assumes that the relation of the two quantities (here, a biomarker of amyloidosis and a biomarker of neurodegeneration) is linear. The pTau/Aß1-42 ratio provides a more reliable interpretation of the AD biomarkers when the Aβ1-42/1-40 ratio is not available, particularly in clear cases. However, it should be considered a purely mechanistic concept.“ lines 367-375

Comments 9: In the CSF immunoassay, what is the ratio of Tau/pTau level? Does it indicate any relation with dementia which can be linked with ERS for combined prediction?

Response 9: Phospho-Tau181 and tTau levels are highly correlated across all CSF biomarker constellations and therefore not recommended for dementia risk prediction, see for example Oberstein, T. J., Schmidt, M. A., Florvaag, A., Haas, A. L., Siegmann, E. M., Olm, P., ... & Maler, J. M. (2022). Amyloid-β levels and cognitive trajectories in non-demented pTau181-positive subjects without amyloidopathy. Brain, 145(11), 4032-4041.

Comments 10: All the abbreviations (AIC, AUC, MMSE, etc) should be mentioned first when the term appears in the manuscript and not randomly elsewhere

Response 10: Thank you for pointing this out, we have corrected it.

Comments 11:Since the samples were obtained from clinical studies the PET imaging data must be available for most of the samples. Can that data be correlated with the ERS score or Aβ 1-42/1-40 ratio?

Response 11: As FDG-PET was not performed consistently in all patients and does not have the diagnostic quality of amyloid PET or CSF findings, we did not include it in the data analysis.